# Long-distance decay-less spin transport in indirect excitons in a van der Waals heterostructure

Zhiwen Zhou ⬭, E. A. Szwed, D. J. Choksy, L. H. Fowler-Gerace & L. V. Butov ⬭ ✉

In addition to its fundamental interest, the long-distance spin transport is essential for spintronic devices. However, the spin relaxation caused by scattering of the particles carrying the spin limits spin transport. We explored spatially indirect excitons (IXs) in van der Waals heterostructures composed of atomically thin layers of transition-metal dichalcogenides as spin carries. We observed the long-distance spin transport: the spin polarized excitons travel over the entire sample, ~10 micron away from the excitation spot, with no spin density decay. This transport is characterized by the $1/e$ decay distances reaching ~100 micron. The $1/e$ decay distances are extracted from fits over the ~10 micron sample size. The emergence of long-distance spin transport is observed at the densities and temperatures where the IX transport decay distances and, in turn, scattering times are strongly enhanced. The suppression of IX scattering suppresses the spin relaxation and enables the long-distance spin transport.

The physics of spin transport includes a number of fundamental phenomena, such as the current-induced spin orientation (the spin Hall effect)[1–4], the spin drift, diffusion and drag[5–7], the quantum spin Hall effect[8–10], and the persistent spin helix[11]. In addition to its fundamental interest, long-distance spin transport with suppressed spin losses is essential for developing spintronic devices, which may offer advantages in dissipation, size, and speed over charge-based devices[12].

Spatially indirect excitons (IXs), also known as interlayer excitons, in heterostructures (HS) can enable the realization of the long-distance spin transfer. IXs are composed of electrons and holes confined in separated layers[13]. Due to the separation of electron and hole layers, the IX lifetimes can exceed the lifetimes of spatially direct excitons (DXs) by orders of magnitude. Due to their long lifetimes, IXs can cool down below the temperature of quantum degeneracy and form a condensate[14] and can travel over long distances[15]. Traveling particles can transfer the spin state. However, the particle scattering causes fluctuating effective magnetic fields originating from the spin-orbit interaction in noncentrosymmetric materials and, as a result, causes the spin relaxation that limits the spin transfer[16]. Therefore, the range of spin transport can be extended by suppressing the scattering of the particles carrying the spin states. This can be achieved with IXs: The suppression of scattering in IX condensate can suppress the spin

relaxation and allow long-distance spin transport. In addition, in contrast to DXs, the electron-hole separation in IXs suppresses the overlap of the electron and hole wave functions and, as a result, suppresses the spin relaxation due to electron-hole exchange[17].

IXs can be created in various HS, in particular, in GaAs HS[18–22], in GaN HS[23], and in ZnO HS[24]. Since the temperature of quantum degeneracy, which can be achieved for excitons, scales proportionally to the exciton binding energy $E_X$[25], IXs with high $E_X$ can form a platform for the realization of high-temperature long-distance spin transport.

IXs in GaAs HS have low $E_X \lesssim 10$ meV[26,27], and the highest $E_X \sim 30$ meV for IXs in III–V and II–VI semiconductor HS is achieved in ZnO HS[24]. Van der Waals HS composed of atomically thin layers of transition-metal dichalcogenides (TMD) enable the realization of excitons with remarkably high binding energies[28–31] and $E_X$ for IXs in TMD HS reach hundreds of meV[25,32,33].

TMD HS also give an opportunity to explore spin transport in periodic potentials due to moiré superlattices. The period of the latter $b \approx a/\sqrt{\delta\theta^2 + \delta^2}$ is typically in the 10 nm range ($a$ is the lattice constant, $\delta$ the lattice mismatch, $\delta\theta$ the deviation of the twist angle between the layers from $i\pi/3$, $i$ is an integer)[34–49]. The moiré potentials can be affected by atomic reconstruction[50–52] and by the disorder.

Department of Physics, University of California San Diego, La Jolla, CA, USA. ✉e-mail: lvbutov@physics.ucsd.edu

In addition, due to the coupling of the spin and valley indices in TMD HS[53–56], the spin transport is coupled to the valley transport (therefore, for simplicity, we will use the term 'spin' also for 'spin-valley').

Detecting the transport of spin-polarized excitons via spatially- and polarization-resolved imaging of exciton luminescence gives the direct measurement of spin transport. Earlier studies using this method led to the observation of spin transport with $1/e$ decay distances $d_{1/e}^{s}$ up to a few μm in IXs in TMD HS[57–60]. Spin transport with $d_{1/e}^{s}$ of a few $\mu m$ was also observed in DXs[61], and the excitation-induced polarization was found to lead to the emergence of ferromagnetic order[62] and to electron or hole spin transport with a spin diffusion length up to ca. 20 μm[63,64] in TMD. Spin relaxation due to scattering of the particles carrying the spin-limited spin transport distances[16].

In this work, we observed in a MoSe$_2$/WSe$_2$ HS the IX mediated long-distance spin transport: the spin polarized excitons travel over the entire sample, ~10 micron away from the excitation spot, with no spin density decay. This transport is characterized by the $1/e$ decay distances reaching ~100 μm. The long-distance spin transport vanishes at high temperatures. With increasing IX density, we observed spin localization, then long-distance spin transport, and then reentrant spin localization.

## Results

### MoSe$_2$/WSe$_2$ heterostructure

We study MoSe$_2$/WSe$_2$ HS assembled by stacking mechanically exfoliated 2D crystals [Supplementary Fig. S1]. IXs are formed from electrons and holes confined in adjacent MoSe$_2$ and WSe$_2$ monolayers (ML), respectively, encapsulated by hBN layers. No voltage is applied in the HS. IXs form the lowest-energy exciton state in the MoSe$_2$/WSe$_2$ HS (Supplementary Fig. S1). The HS details are presented in SI.

### IX generation and detection

Both the long-distance IX transport[65] and the long-distance spin transport, which is described below, are realized when the optical excitation has the energy $E_{ex}$ close to the energy of DXs in the HS. The resonant excitation allows for lowering the excitation-induced heating of the IX system. In particular, the colder IXs created by the resonant excitation screen the HS disorder more effectively[66–68]. In this work, the laser excitation with $E_{ex} = 1.689$ eV is resonant to WSe$_2$ DX.

Both spin generation and detection in IXs are achieved by optical means via photon polarization. The circularly polarized laser excitation is focused to a ~2 μm spot, and the spin propagation is detected by the polarization-resolved PL imaging. Figure 1a shows a high degree of circular polarization in the excitation spot, indicating an effective transfer of the optically generated spin-polarized DXs to the spin-polarized IXs with the spin relaxation time long compared to the exciton recombination and energy relaxation times.

### IX-mediated spin transport

The propagation of spin-polarized IXs from the excitation spot transfer the spin polarization. For the particular IX densities and temperatures,

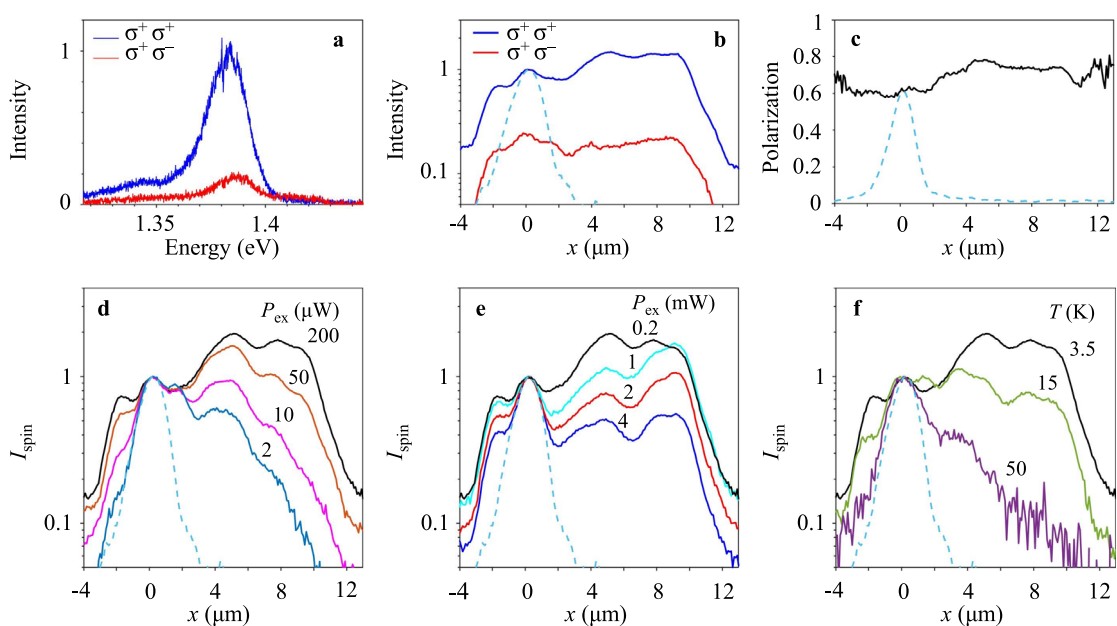

**Fig. 1 | The long-distance spin-valley transport in IXs in MoSe$_2$/WSe$_2$ HS. a** The circular polarization of IX PL. The blue (red) spectrum is co-polarized (cross-polarized) with the circularly polarized laser excitation. **b** Co-polarized $I_{\sigma^+}$ (blue) and cross-polarized $I_{\sigma^-}$ (red) IX PL intensity vs. the distance from the laser excitation spot centered at $x = 0$. The HS active area extends from $x \sim -3$ to 10 μm. The polarized IX PL propagates through the entire HS. **c** The degree of circular polarization of IX PL $P = (I_{\sigma^+} - I_{\sigma^-})/(I_{\sigma^+} + I_{\sigma^-})$ vs. the transport distance. No decay is observed for the polarization transport for IXs over the entire HS. The laser excitation power $P_{ex} = 0.2$ mW, $T = 1.7$ K (**a–c**). **d–f** Normalized spin density profiles $I_{spin} = I_{\sigma^+} - I_{\sigma^-}$ for the LE-IXs for different $P_{ex}$ (**d**, **e**) and temperatures (**f**). In (**d**),

$P_{ex} = 2, 10, 50, 200$ μW (bottom to top). In (**e**), $P_{ex} = 0.2, 1, 2, 4$ mW (top to bottom). In (**c**), $T = 3.5, 15, 50$ K (top to bottom). The spin density transport non-monotonically varies with increasing $P_{ex}$, increases at $P_{ex} \lesssim 0.2$ mW (**d**) and reduces at $P_{ex} \gtrsim 0.2$ mW (**e**), and vanishes at high temperatures (**f**). The dashed line in (**b–f**) shows the DX luminescence profile in the MoSe$_2$ ML, this profile is close to the laser excitation profile for short-range DX transport. The LE-IX spectra are separated from the HE-IX spectra by the spectral integration in the range $E < 1.4$ eV (**b–f**). The HE-IXs appear in the spectra at high $P_{ex} \gtrsim 0.2$ mW (Fig. 3). $T = 3.5$ K (**d,e**), $P_{ex} = 0.2$ mW (**f**). The ~2 μm laser spot is centered at $x = 0$ (**a–f**).

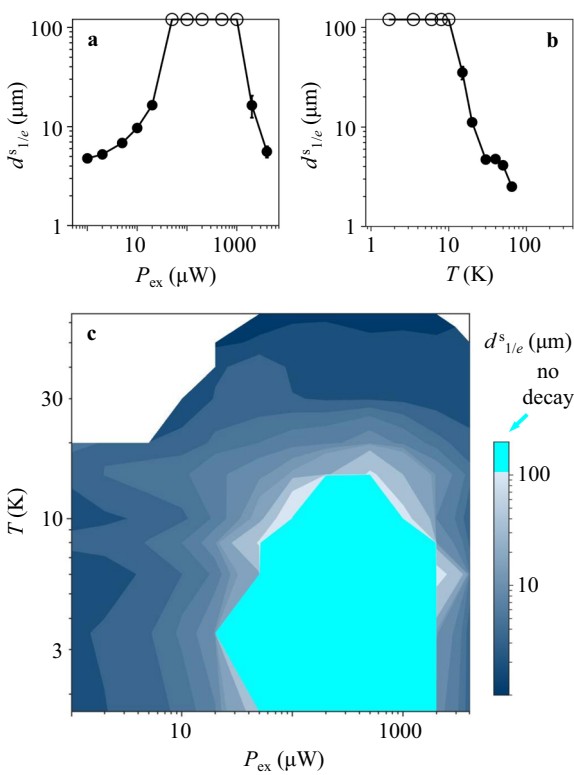

**Fig. 2 | Excitation power and temperature dependence of spin transport in LE-IXs. a–c** The $1/e$ decay distance $d^s_{1/e}$ of spin density transport $I_{\text{spin}} = I_{\sigma^+} - I_{\sigma^-}$ in LE-IXs vs. the laser excitation power $P_{\text{ex}}$ (**a**), vs. temperature (**b**), and vs. $P_{\text{ex}}$ and temperature (**c**). $d^s_{1/e}$ are obtained from least-squares fitting the LE-IX spin density transport profiles $I_{\text{spin}}(x)$ (Fig. 1d–f) to exponential decays in the region from the excitation spot to the HS edge, $x = 0 - 9\ \mu\text{m}$. The data with the fit indicating diverging $d^s_{1/e}$ are presented by circles on the edge (**a**, **b**) or by cyan color (**c**). The error bars represent the uncertainty in least-squares fitting the spin transport decays to exponential decays. The LE-IX spectra are separated from the HE-IX spectra by the Gaussian fits. The HE-IXs appear in the spectra at high $P_{\text{ex}} \gtrsim 0.2\ \text{mW}$ (Fig. 3). $T = 3.5\ \text{K}$ (**a**), $P_{\text{ex}} = 0.2\ \text{mW}$ (**b**).

outlined below, both the intensities of co-polarized and cross-polarized IX PL $I_{\sigma^+}$ and $I_{\sigma^-}$ (Fig. 1b) and the degree of circular polarization of IX PL $P = (I_{\sigma^+} - I_{\sigma^-})/(I_{\sigma^+} + I_{\sigma^-})$ (Fig. 1c) propagate over the entire HS with no losses.

IX transport is characterized by the propagation of total IX intensity in both circular polarizations $n \sim I_{\sigma^+} + I_{\sigma^-}$. In turn, the transport of spin polarization density carried by IXs is characterized by the propagation of $I_{\text{spin}} = Pn = I_{\sigma^+} - I_{\sigma^-}$. The dependence of spin density transport on excitation power $P_{\text{ex}}$ and temperature is described below. The spin transport nonmonotonically varies with increasing $P_{\text{ex}}$, increases at $P_{\text{ex}} \lesssim 0.2\ \text{mW}$ (Fig. 1d) and reduces at $P_{\text{ex}} \gtrsim 0.2\ \text{mW}$ (Fig. 1e), and vanishes at high temperatures (Fig. 1f). The spin transport is characterized by the $1/e$ decay distance of the spin polarization density $d^s_{1/e}$. The latter is obtained from least-squares fitting the spin density transport profiles $I_{\text{spin}}(x)$ (Fig. 1d–f) to exponential decays in the region from the excitation spot to the HS edge. The variation of spin transport with excitation power and temperature is presented by the variation of $d^s_{1/e}$ in Fig. 2. The HS dimensions allow establishing that the longest $d^s_{1/e}$ reach 100 μm (as outlined in Supplementary Note 6). The data with the fit indicating diverging $d^s_{1/e}$, that is, with no spin density decay within the entire HS, are presented by circles on the edge in Fig. 2a, b, and by cyan color in Fig. 2c.

For low $P_{\text{ex}}$, a single IX PL line is observed in the spectra. However, a higher-energy IX PL line appears in the spectrum at high $P_{\text{ex}}$ (Fig. 3).

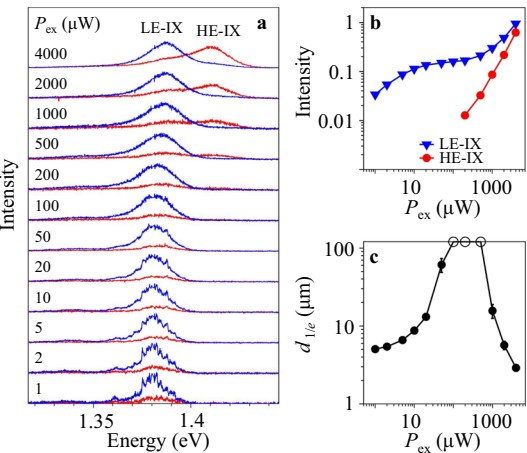

**Fig. 3 | Density dependence of IX PL spectra. a** The excitation power $P_{\text{ex}}$ dependence of co-polarized (blue) and cross-polarized (red) IX spectra. The lower-energy IX (LE-IX) PL is co-polarized. The higher-energy IX (HE-IX) PL is cross-polarized. The HE-IXs appear in the spectra at high $P_{\text{ex}} \gtrsim 0.2\ \text{mW}$. The spectral profile separation of LE-IXs and HE-IXs is presented in Supplementary Fig. S3. **b** The intensity of LE-IX PL (blue triangles) and HE-IX PL (red points) vs. $P_{\text{ex}}$. **c** The $1/e$ LE-IX transport decay distance $d_{1/e}$ vs. $P_{\text{ex}}$. $d_{1/e}$ are obtained from least-squares fitting the spectrally integrated LE-IX PL intensity including both polarizations $I_{\sigma^+} + I_{\sigma^-}$ to exponential decays in the region from the excitation spot to the HS edge, $x = 0 - 9\ \mu\text{m}$. The data with the fit indicating diverging $d_{1/e}$ are presented by circles on the edge. The error bars represent the uncertainty in least-squares fitting the LE-IX transport decays to exponential decays. The appearance of HE-IX in the spectrum (**a**, **b**) correlates with the onset of IX transport suppression (**c**). $T = 3.5\ \text{K}$.

We will refer to the IXs corresponding to these PL lines as the lower-energy IXs (LE-IXs) and higher-energy IXs (HE-IXs). Figures 1 and 2 present the spin transport carried by LE-IXs.

## Discussion
### Comparison of spin transport and IX transport
The data are discussed below. The spin transport (Fig. 2) is carried by LE-IXs and can be compared with the LE-IX transport (Fig. 3c and Supplementary Fig. S7). Due to the separation $d_z$ between the electron and hole layers, IXs have electric dipoles $ed_z$, and the interaction between IXs is repulsive[69]. IXs in moiré superlattices form a system of repulsively interacting bosons in periodic potentials. The enhancement followed by the suppression of the LE-IX transport with density (Fig. 3c) is in qualitative agreement with the Bose-Hubbard theory of bosons in periodic potentials predicting superfluid at $N \sim 1/2$ and insulating at $N \sim 0$ and $N \sim 1$ phases for the number of bosons per site of the periodic potential $N$[70]. For the maximum LE-IX transport distances observed at $P_{\text{ex}} \sim 0.2\ \text{mW}$ (Fig. 3c), the LE-IX density $n$ estimated from the energy shift $\delta E$ as $n = \delta E \varepsilon/(4\pi e^2 d_z)$[69] is $n \sim 2 \times 10^{11}\ \text{cm}^{-2}$ ($d_z \sim 0.6\ \text{nm}$, the dielectric constant $\varepsilon \sim 7.4$[71]). This density is well below the Mott transition density $n_{\text{Mott}} > 10^{12}\ \text{cm}^{-2}$[25,72]. $N \sim 1/2$ at $n \sim 2 \times 10^{11}\ \text{cm}^{-2}$ for the moiré superlattice period $b = 17\ \text{nm}$. This period $b \sim a/\delta\theta$ corresponds to the twist angle $\delta\theta = 1.1°$, which agrees with the angle between $MoSe_2$ and $WSe_2$ edges in the HS (Supplementary Fig. S1). This rough estimate indicates that the observation of LE-IX localization, then long-range transport, and then localization with increasing density (Fig. 3c) is in agreement with the Bose-Hubbard theory[70].

In contrast, the data disagree with the classical diffusive transport. For classical transport, a substantial increase of transport distance with density occurs when the IX interaction energy becomes comparable to the amplitude of in-plane potential so it can be screened by the repulsively interacting IXs[68]. However, the amplitude of in-plane disorder and moiré potential in $MoSe_2$/$WSe_2$ HS, tens of meV[34–49], is

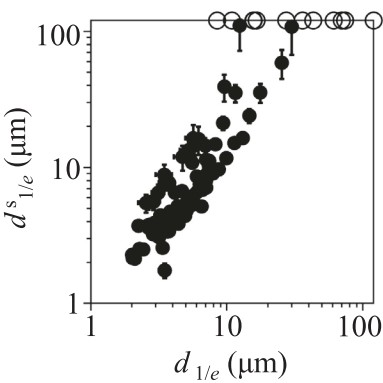

**Fig. 4 | Correlation between the spin transport 1/e decay distances $d^s_{1/e}$ and the LE-IX transport 1/e decay distances $d_{1/e}$.** The values for $d^s_{1/e}$ and $d_{1/e}$ are taken from Fig. 2 and Supplementary Fig. S7. The data with the fit indicating diverging $d^s_{1/e}$ and $d_{1/e}$ are presented by circles on the edge. The error bars represent the uncertainty in least-squares fitting the transport decays to exponential decays. The enhancement of $d^s_{1/e}$ with $d_{1/e}$ is observed in a broad range of both excitation density and temperature variations, corresponding to the range of these parameters in Fig. 2 and Supplementary Fig. S7.

significantly higher than the IX interaction energy given by the LE-IX energy shift with density, a few meV (Fig. 3a). Therefore, the IX interaction energy is insufficiently strong to screen the in-plane potential and neither the long-distance decay-less LE-IX transport (Fig. 3c) nor the long-distance decay-less spin transport (Fig. 2) can occur due to classical screening of the in-plane potential[68]. In addition, the transport suppression at high densities (Fig. 3c) is inconsistent with the classical diffusive transport, which enhances with density[58,60,68]. Furthermore, the long-range transport vanishes at high temperatures (Supplementary Fig. S7), which is inconsistent with the classical diffusive transport, which enhances with temperature[58,60,68].

High values of $d_{1/e}$ (Fig. 3c) indicate high values of IX diffusivity. In turn, diffusivities are proportional to scattering times, and high IX diffusivities indicate suppression of IX scattering. For instance, for classical diffusive LE-IX transport, the IX diffusivity and mean free time (scattering time) are given by $\sim (d_{1/e})^2/\tau$ and $\sim (d_{1/e})^2/\tau \cdot m/(k_B T)$, respectively ($\tau$ and $m$ are the IX lifetime and mass, $m$ ~ free electron mass and $\tau$ ~ 10 ns for the HS[65]). These values become anomalously high for $d_{1/e}$ ~ 100 μm. However, as outlined above, the long-range IX transport is beyond classical diffusive transport, and, therefore, accurate estimates of IX diffusivities and scattering times should go beyond the formulas for classical diffusion and that is the subject of future works.

The density dependence of spin transport (Fig. 2a) is qualitatively similar to the density dependence of LE-IX transport (Fig. 3c): Both spin transport and LE-IX transport first enhance and then suppress with density. The temperature dependence of spin transport is qualitatively similar to the temperature dependence of LE-IX transport: Both the long-distance spin transport (Fig. 2b) and the long-distance LE-IX transport (Supplementary Fig. S7) vanish at $T$ ~ 10 K. The parameters for the long-distance decay-less spin transport (Fig. 2) correlate with the parameters for the long-range LE-IX transport (Fig. 3c and Supplementary Fig. S7). Figure 4 shows that the enhancement of $d^s_{1/e}$ with $d_{1/e}$ is observed in a broad range of both excitation density and temperature variations, corresponding to the range of these parameters in Fig. 2 and Supplementary Fig. S7. The correlation of $d^s_{1/e}$ with $d_{1/e}$ and, in turn, enhanced IX scattering time suggests the suppression of scattering as the mechanism of the long-distance decay-less spin transport. This complies with the scattering being the mechanism of spin relaxation that limits the spin transfer[16].

The enhanced $d_{1/e}$ and, in turn, scattering time is observed in the range of temperatures and densities consistent with those predicted

for superfluidity by the Bose-Hubbard theory[70] as outlined above. Therefore, superfluidity can be the origin of the enhanced scattering time.

The long-distance decay-less spin transport vanishes at ~10 K (Fig. 2b, c). The mechanism of suppression of spin relaxation due to suppression of scattering of IXs carrying the spin indicates that long-distance spin transport with suppressed losses can be achieved at high temperatures in IX systems with high superfluidity temperatures $T_c$. The theory predicts that the superfluidity temperature for bosons in periodic potentials $T_c$ ~ $4\pi NJ$ and higher $T_c$ can be achieved in lattices with higher inter-site hopping $J$[73]. Higher $J$ can be achieved in moiré superlattices with smaller periods in HS with larger twist angles $\delta\theta$, or in moiré superlattices with smaller amplitudes that can be realized in HS with the same-TMD electron and hole layers[74,75], or by lowering the moiré superlattice amplitude by voltage[37,76], or by adding a spacer (hBN) layer between the electron and hole layers[58]. For TMD HS with suppressed moiré potentials, the theory predicts high-$T_c$ superfluidity[25,32]. This, in turn, can enable the realization of high-temperature long-distance spin transport with suppressed losses.

### Lower-energy and higher-energy IXs

The above data outline the long-distance spin transport carried by LE-IXs. Figure 3 shows that HE-IX PL appears in the spectrum at high $P_{ex}$. In contrast to the LE-IX PL, which is co-polarized, the HE-IX PL is cross-polarized with circularly polarized laser excitation. A similar higher-energy IX PL was observed in earlier studies. Various interpretations for multiple IX PL lines were considered, including the excitonic states split due to the conduction band K-valley spin splitting[77], trions[78], excitonic states indirect in momentum space and split due to the valley energy difference[79,80] or spin-orbit coupling[81], excitonic states in moiré superlattices[40–46] and, recently, excitonic states in moiré lattice sites with single and double occupancy[82,83]. Our data show that the appearance of HE-IX PL in the spectrum (Fig. 3a, b) correlates with the onset of IX reentrant localization (Fig. 3c). As outlined above, in the regime of reentrant localization, the occupation of moiré cells becomes high. This results in the appearance of moiré cells with double occupancy. Therefore, the IX reentrant localization in transport measurements (Fig. 3c) is consistent with the appearance of high-energy IX PL in the spectra (refs. 82,83 and Fig. 3a, b). The intra-cell IX repulsion enhances the IX energy. This is in qualitative agreement with a higher HE-IX energy (Fig. 3a). (Spin transport in HE-IXs is shorter-range than in LE-IXs and is not considered in this work.) (Narrow PL lines at low densities (Fig. 3a) can be related to localized states and are not considered in this work.)

In summary, we observed in a MoSe₂/WSe₂ HS the IX-mediated long-distance spin transport: the spin-polarized IXs travel over the entire sample, ~10 μm away from the excitation spot, with no spin density decay. This transport is characterized by the 1/e decay distances reaching ~100 μm. The emergence of long-distance spin transport is observed at the densities and temperatures where the IX transport decay distances and, in turn, scattering times are strongly enhanced. The suppression of IX scattering suppresses the spin relaxation and enables long-distance spin transport. This mechanism of protection against spin relaxation makes IXs a platform for the realization of long-distance decay-less spin transport.

## Methods
### Van der Waals HS
The MoSe₂/WSe₂ HS was assembled using the dry-transfer peel-and-lift technique[84]. The heterostructure details are presented in Supplementary Note 1.

### Optical measurements
Excitons were generated by a cw Ti:Sapphire laser with the excitation energy $E_{ex}$ = 1.689 eV. PL spectra were measured using a spectrometer with a resolution of 0.2 meV and a liquid-nitrogen-cooled CCD.

The spatial profiles of polarization-resolved IX PL vs. $x$ were obtained from the polarization-resolved PL images detected using the CCD. The signal was integrated from $y = -0.5$ to $y = +0.5\,\mu m$. Representative polarization-resolved IX PL images are presented in Supplementary Note 2.

The experiments were performed in a variable-temperature 4He cryostat. The sample was mounted on an Attocube xyz piezo translation stage allowing adjusting the sample position relative to a focusing lens inside the cryostat. All phenomena presented in this work are reproducible after multiple cooling down to 2 K and warming up to room temperature.

## Data availability
Source data files are available via Figshare at https://doi.org/10.6084/m9.figshare.27003031. All relevant data are available from the corresponding author upon reasonable request.

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

## Acknowledgements

We thank M.M. Fogler and J.R. Leonard for discussions. We especially thank A.H. MacDonald for discussions of IXs in moiré potentials and A.K. Geim for teaching us manufacturing TMD HS. The studies were supported by the DOE Office of Basic Energy Sciences under Award DE-FG02-07ER46449 (L.V.B.). The HS manufacturing was supported by NSF Grant 1905478 (L.V.B.).

## Author contributions

L.V.B. designed the project. L.H.F.-G. manufactured the TMD heterostructure. Z.Z., E.A.S., and D.J.C. performed the measurements. Z.Z. and L.V.B. analyzed the data. L.V.B. wrote the manuscript with inputs from all the authors.

## Competing interests

The authors declare no competing interests.
