## [Transparent Peer Review file · Nature Communications]

Long-distance decay-less spin transport in indirect excitons in a van der Waals heterostructure

Corresponding Author: Professor Leonid Butov

This manuscript has been previously reviewed at another journal. This document only contains reviewer comments, rebuttal and decision letters for versions considered at Nature Communications.

A version of this paper was originally rejected for publication by Nature Communications, however that decision was reconsidered after appeal by the authors.

Version 0:

Reviewer comments:

Reviewer #2

(Remarks to the Author)

Zhu et al unfortunately still insist on their claim of the 100 um decay length which is simply not appropriate. It is not based on an actual experimental observation but by fitting an observed PL profile to a Gaussian curve (when no such curve can be recognized from the data) and extrapolating by a factor of 7 beyond the sample size. Without seeing an actual diffusion over 100 um, it is not possible to make this claim, unfortunately the authors keep repeating it many times over in their manuscript.

In addition, there is a lot of similarity between this manuscript and the recently published paper in Nature photonics doi:10.1038/s41566-024-01435-w by the same authors. Figures 1d,e,f appear to duplicated between the manuscripts.

Reviewer #3

(Remarks to the Author)

The authors responded to all the comments from the reviewers. The paper can be published in the current state.

Version 1:

Reviewer comments:

Reviewer #4

(Remarks to the Author)

The revised manuscript has been improved and the authors now clearly state in the main text and abstract that their extracted 100 micron spin/valley length scale was extracted from a ~10 micron sample. Given that this is clearly stated, and the authors explain how they extracted the value of 100 microns, I am satisfied with the results as they are presented. I do understand the point of view of reviewer 2, and agree that a more conservative and qualitative claim could be made stating that there is "no significant decay over the sample area". Instead, the authors have chosen to perform a quantitative analysis of the IX decay, and their result is ~100 microns. I do agree that this is a "bold" claim, but the authors are transparent about their method and provide adequate discussion of the sample size. They do not claim that this is a direct observation of spin valley transport across 100 microns. More details (about the fitting and error analysis on the 100 micron number) could be presented in the main text to reduce this source of criticism.

The authors have recently published a Nature Photonics paper entitled "Transport and localization of indirect excitons in a van der Waals heterostructure". At first glance the results appear similar, however it is clear that the Nature Photonics paper solely focused on IX population transport, and not spin/valley transport. The similarity comes because it appears that the spin-valley polarization largely tracks with the IX density. It seems clear that these are related results, but the authors made a

decision to write two separate papers. In the Nature Photonics paper, which appears to be studying the same ~10 micron sample, the authors publish a similar analysis and claim ~100 microns for the IX transport decay length. Given that this Nature Photonics work has been published, that can be used as additional validation of the decay length claim since it was reviewed independently.

Response to Remarks of Reviewer 1 in the previous round (Reviewer's comments are marked by italic font, changes in the manuscript are marked by blue font). The Remarks of Reviewer 1 and changes in the manuscript in response to them are essential for understanding the revised manuscript and are presented here.

Reviewer #1 (Remarks to the Author):

The revised manuscript is significantly improved. I am now supportive of publication. In particular, the author's emphasis on the density dependence of the decay length has been made clear which does increase the novelty of the work.

I do strongly suggest that the authors clearly state (even in the abstract) that the ~100 micron decay length scale is extracted from a fit over the ~ 10-12 micron sample size.

We thank Reviewer 1 for the high estimate of our work and useful suggestions. We follow the suggestion of Reviewer 1 and state in the abstract “The 1/e decay distances are extracted from fits over the ~ 10 micron sample size.”

We also state the region of the fit “x = 0 – 9 μm” in the captions to Fig. 2 and Fig. 3 where the decay lengths are presented.

Response to Remarks of Reviewer 2 (Reviewer's comments are marked by italic font, changes in response to remarks of Reviewers in the previous review round, before this report of Reviewer 2, are marked by blue font, new changes – the addition of published Nature Photonics to Ref. 66, are marked by cyan font).

Reviewer #2 (Remarks to the Author):

Zhu et al unfortunately still insist on their claim of the 100 um decay length which is simply not appropriate. It is not based on an actual experimental observation but by fitting an observed PL profile to a Gaussian curve (when no such curve can be recognized from the data) and extrapolating by a factor of 7 beyond the sample size. Without seeing an actual diffusion over 100 um, it is not possible to make this claim, unfortunately the authors keep repeating it many times over in their manuscript.

The 1/e decay length of spin transport in our work does reach 100 um. This is a mathematical fact. The least-squares fit used to obtain the 1/e decay length is unambiguous math. This math gives 100 um and we cannot change that. All we can do is to describe how the fit is done and we accurately do this in the manuscript, starting from the abstract as recommended by Reviewer 1: “the spin polarized excitons travel over the entire sample, ~ 10 micron away from the excitation spot, with no spin density decay. This transport is characterized by the 1/e decay distances reaching 100 micron. The 1/e decay distances are extracted from fits over the ~ 10 micron sample size.” The fit curves are shown on Fig. S10 and the details, including the accuracy of the fits, are carefully described in Section “Estimates of \$d_{1/e}^s\$ ” in SI.

Obviously, we do not claim “*a Gaussian curve*” and we do not claim IX transport over 100 um. In contrast, we show that spin transport found in our work is significantly longer than diffusive spin transport in all earlier studies. This is clear just from the measured spin density profiles shown in Fig. A, which compares spin transport in our work with spin transport in earlier works on IXs in TMD. Figure A also shows that at high temperatures ($T > 15$ K), where the long-distance spin transport vanishes in our experiments, the spin transport in our work become comparable to spin transport in earlier studies.

[Redacted]

To quantify the variation of spin transport with excitation density and temperature, we introduce the $1/e$ spin transport decay distance $d^{1/e}$. This parameter shows the variation of spin transport with excitation density and temperature (Fig. 2). We unambiguously describe how this parameter is defined already in the abstract as recommended by Reviewer 1 and outlined above.

In addition, there is a lot of similarity between this manuscript and the recently published paper in Nature photonics doi:10.1038/s41566-024-01435-w by the same authors. Figures 1d,e,f appear to duplicated between the manuscripts.

The criticism of Reviewer 2 "*Figures 1d,e,f appear to duplicated between the manuscripts*" is a factual error because Figures 1d,e,f in our manuscript show spin transport, while, in contrast, all figures in the other manuscript arXiv:2307.00702 (Nature Photonics) present IX transport rather than spin transport.

Our manuscript shows that the difference between the long-distance spin transport found in our work and the diffusive spin transport in earlier works is dramatic. To identify the mechanism governing the extraordinary spin transport, which finding is presented in our manuscript, we compared IX transport in Ref. [66] (arXiv:2307.00702, Nature Photonics) with spin transport in this manuscript. This comparison identified the new mechanism of protection against the spin relaxation in TMD: "The suppression of IX scattering suppresses the spin relaxation and enables the long-distance spin transport".

When we compare spin transport with IX transport, as required to identify the origin of long-distance spin transport, we carefully differentiate our results on spin transport presented in this manuscript from our results on IX transport presented in Ref. [66] (arXiv:2307.00702, Nature Photonics). To avoid any possibility of accusation of hiding similarity in findings by difference in description styles, we used similar style to describe spin transport and IX transport in these two works. The similar description style emphasizes the difference in the phenomena: spin transport is different from IX transport. In particular, this is clear in the following examples:

- **This manuscript:** "We observed the long-distance spin transport: the spin polarized excitons travel over the entire sample, ~ 10 micron away from the excitation spot, with no spin density decay. This transport is characterized by the $1/e$ decay distances reaching ~ 100 micron." "The long-distance decay-less spin transport vanishes at ~ 10 K"

Nature Photonics: "Here, in a MoSe₂/WSe₂ heterostructure, we present the IX long-range transport with $1/e$ decay distances reaching and exceeding $100 \mu\text{m}$. The IX long-range transport vanishes at temperatures above ~ 10 K."

- **This manuscript:** “With increasing IX density, we observed spin localization, then long-distance spin transport, and then reentrant spin localization. The nonmonotonic density dependence is in qualitative agreement with the Bose-Hubbard theory prediction for superfluid and insulating phases in periodic potentials due to moiré superlattices.”

Nature Photonics: “With increasing IX density, IX localization followed by IX long-range transport and IX re-entrant localization are observed. The non-monotonic dependence of IX transport on density is in qualitative agreement with the Bose-Hubbard theory prediction for superfluid and insulating phases in periodic potentials of moiré superlattices”.

Keeping the same description style emphasizes that these sentences present two different phenomena - spin transport in this manuscript and IX transport in Ref. [66] (arXiv:2307.00702, Nature Photonics).

When comparing the long-range spin transport, which finding is presented in this manuscript, with the long-range IX transport, which finding is presented in Ref. [66] (arXiv:2307.00702, Nature Photonics), we properly reference Ref. [66] (arXiv:2307.00702, Nature Photonics). This comparison is required to show that the found spin transport is not diffusive and, in contrast, is governed by the new mechanism: “The suppression of IX scattering suppresses the spin relaxation and enables the long-distance spin transport”, as described in the manuscript. Nature Photonics paper was published during the manuscript review and we add a reference to it in Ref [66]: “L.H. Fowler-Gerace, Zhiwen Zhou, E.A. Szwed, D.J. Choksy, L.V. Butov, Transport and localization of indirect excitons in a van der Waals heterostructure. arXiv:2307.00702 (2023); Nat. Photonics 18, 823 (2024).”

Response to Remarks of Reviewer 3 (Reviewer’s comments are marked by italic font, changes in response to remarks of Reviewers in the previous review round, before this report of Reviewer 3, are marked by blue font in the manuscript).

Reviewer #3 (Remarks to the Author):

The authors responded to all the comments from the reviewers. The paper can be published in the current state.

We thank Reviewer 3 for the high estimate of our work.

Response to REVIEWERS' COMMENTS (Reviewer's comments are marked by italic font).

Reviewer #4 (Remarks to the Author):

The revised manuscript has been improved and the authors now clearly state in the main text and abstract that their extracted 100 micron spin/valley length scale was extracted from a ~10 micron sample. Given that this is clearly stated, and the authors explain how they extracted the value of 100 microns, I am satisfied with the results as they are presented.

I do understand the point of view of reviewer 2, and agree that a more conservative and qualitative claim could be made stating that there is "no significant decay over the sample area". Instead, the authors have chosen to perform a quantitative analysis of the IX decay, and their result is ~100 microns. I do agree that this is a "bold" claim, but the authors are transparent about their method and provide adequate discussion of the sample size. They do not claim that this is a direct observation of spin valley transport across 100 microns. More details (about the fitting and error analysis on the 100 micron number) could be presented in the main text to reduce this source of criticism.

We state that "there is "no significant decay over the sample area"": "the spin polarized excitons travel over the entire sample, ~ 10 micron away from the excitation spot, with no spin density decay". In addition to this, we "perform a quantitative analysis of the IX decay", in particular, because we found strong variations of spin transport with density and temperature and describing these variations requires "a quantitative analysis", done using the $1/e$ decay distances. The sample size is explicitly stated in the manuscript and shown in Fig. S1 and, obviously, IXs do not go beyond the sample as shown in Fig. 1d-f. Therefore, the reader will not confuse the "spin valley transport across 100 microns" and the $1/e$ decay distances ~ 100 microns.

For the analysis, we use standard least-squares fitting: "The spin transport is characterized by the $1/e$ decay distance of the spin polarization density $d_{1/e}^s$. The latter are obtained from least-squares fitting the spin density transport profiles $I_{\text{spin}}(x)$ (Fig. 1d-f) to exponential decays in the region from the excitation spot to the HS edge." The details, are described in Supplementary Note 6 in SI and, following the suggestion of Reviewer 4, we add a reference to this description in the main text: "The HS dimensions allow establishing that the longest $d_{1/e}^s$ reach 100 μm (as outlined in Supplementary Note 6 in SI)".

The authors have recently published a Nature Photonics paper entitled "Transport and localization of indirect excitons in a van der Waals heterostructure". At first glance the results appear similar, however it is clear that the Nature Photonics paper solely focused on IX population transport, and not spin/valley transport. The similarity comes because it appears that the spin-valley polarization largely tracks with the IX density. It seems clear that these are related results, but the authors made a decision to write two separate papers. In the Nature Photonics paper, which appears to be studying the same ~10 micron sample, the authors publish a similar analysis and claim ~100 microns for the IX transport decay length. Given that this Nature Photonics work has been published, that can be used as additional validation of the decay length claim since it was reviewed independently.

To identify the mechanism governing the long-range spin transport, which finding is presented in our manuscript, we compare IX transport presented in our Nature Photonics paper (Ref. 66) with spin transport in this manuscript. This comparison identifies the mechanism of protection against the spin relaxation: "The suppression of IX scattering suppresses the spin relaxation and enables the long-distance spin transport".

When we compare spin transport with IX transport, we differentiate our results on spin transport presented in this manuscript from our results on IX transport presented in Ref. [66] and properly reference Ref. [66]. To avoid any possibility of accusation of hiding similarity in findings by difference in description styles, we use similar style to describe spin transport and IX transport in these two works. The similarity in the description style emphasizes the difference in the phenomena: spin transport is different from particle transport. We believe the readers will agree.